# Identification of TAT as a Biomarker Involved in Cell Cycle and DNA Repair in Breast Cancer

**DOI:** 10.3390/biom14091088

**Published:** 2024-08-30

**Authors:** Fei Xie, Saiwei Hua, Yajuan Guo, Taoyuan Wang, Changliang Shan, Lianwen Zhang, Tao He

**Affiliations:** 1State Key Laboratory of Medicinal Chemical Biology, College of Pharmacy and Tianjin Key Laboratory of Molecular Drug Research, Nankai University, Tianjin 300350, China; 1120200602@mail.nankai.edu.cn (F.X.); 2120231670@mail.nankai.edu.cn (S.H.); 2120221590@mail.nankai.edu.cn (Y.G.); changliangshan@nankai.edu.cn (C.S.); 2Cardiothoracic Surgery Department, Characteristic Medical Center of the Chinese People’s Armed Police Force, Tianjin 300162, China; wangtaoyuan@wj120.cn; 3Department of Pathology, Characteristic Medical Center of The Chinese People’s Armed Police Force, Tianjin 300162, China

**Keywords:** breast cancer, tumor biomarker, tyrosine aminotransferase, DNA methyltransferase 3, cell cycle, DNA repair

## Abstract

Breast cancer (BC) is the most frequently diagnosed cancer and the primary cause of cancer-related mortality in women. Treatment of triple-negative breast cancer (TNBC) remains particularly challenging due to its resistance to chemotherapy and poor prognosis. Extensive research efforts in BC screening and therapy have improved clinical outcomes for BC patients. Therefore, identifying reliable biomarkers for TNBC is of great clinical importance. Here, we found that tyrosine aminotransferase (TAT) expression was significantly reduced in BC and strongly correlated with the poor prognosis of BC patients, which distinguished BC patients from normal individuals, indicating that TAT is a valuable biomarker for early BC diagnosis. Mechanistically, we uncovered that methylation of the *TAT* promoter was significantly increased by DNA methyltransferase 3 (DNMT3A/3B). In addition, reduced TAT contributes to DNA replication and cell cycle activation by regulating homologous recombination repair and mismatch repair to ensure genomic stability, which may be one of the reasons for TNBC resistance to chemotherapy. Furthermore, we demonstrated that Diazinon increases *TAT* expression as an inhibitor of DNMT3A/3B and inhibits the growth of BC by blocking downstream pathways. Taken together, we revealed that *TAT* is silenced by DNMT3A/3B in BC, especially in TNBC, which promotes the proliferation of tumor cells by supporting DNA replication, activating cell cycle, and enhancing DNA damage repair. These results provide fresh insights and a theoretical foundation for the clinical diagnosis and treatment of BC.

## 1. Introduction

Breast cancer (BC) is the most prevalent cancer in women and is associated with a significant mortality rate [1]. It is categorized into various subtypes based on genetic analysis or immunohistochemical analysis. The subtype lacking estrogen receptor (ER), progesterone receptor (PR), and human epidermal growth factor receptor 2 (HER2) is classified as triple-negative breast cancer (TNBC), accounting for approximately 20% of BCs. Compared with other subtypes, TNBC is notably aggressive and metastatic due to receptor absence, and displays higher resistance to chemotherapy and targeted therapy [2]. Early screening, diagnosis, and typing could effectively reduce BC morbidity and mortality [3,4]. Therefore, the identification of effective and reliable tumor biomarkers has important clinical significance for the early screening, diagnosis, and treatment of BC.

Characteristics of tumors are continually being added [5], in which metabolic reprogramming remains a dominant feature. Glucose metabolism, amino acid metabolism, and fatty acid metabolism/amino acid metabolism in BC have been identified as critical factors in the metastasis and chemoresistance process of BC [6,7]. In addition, BC patients with up-regulated essential amino acid metabolism are more likely to exhibit treatment resistance [8]. These findings suggest a promising approach to explore BC biomarkers from the perspective of amino acid metabolism. The tyrosine-metabolizing enzyme tyrosine aminotransferase (TAT) was previously identified as a BC biomarker by a new computational method [9], but the biological function and molecular mechanism of TAT in BC have not been further explored.

In addition to diagnosis, treating triple-negative breast cancer (TNBC) is another challenge. Studies have shown that DNA methyltransferases are abnormally highly expressed in BC, are closely related to its occurrence and development, and serve as an independent prognostic factor for BC patients [10,11]. Moreover, DNA methyltransferases are also used as drug targets to inhibit BC [12,13,14,15]. Regrettably, the functional mechanisms of DNA methyltransferases in BC have not yet been elucidated.

In this study, we not only obtained similar results as previously reported [9], based on expression differences, but also re-identified TAT as a promising biomarker for BC based on ROC curve analysis and survival prognosis. More importantly, we further revealed the molecular mechanism that the promoter of *TAT* in BC is methylated by DNA methyltransferases (DNMT3A and DNMT3B), resulting in a decrease in *TAT* expression. And the reduced TAT not only contributes to DNA replication, activates the cell cycle, and promotes the progression of BC but also improves DNA damage repair capabilities to ensure orderly DNA replication and maintain genome stability. Furthermore, we identified that Diazinon binds to DNA methyltransferases (DNMT3A/3B) and inhibits their activity, thereby playing a tumor suppressor role by increasing the expression of *TAT* in BC. In conclusion, the results of this study indicate that TAT is repressed by epigenetic regulation in BC, and Diazinon activates TAT by inhibiting DNMT3A/3B, thereby promoting the occurrence and development of BC. These findings provide valuable insights into the molecular typing of BC and the development of new treatment strategies in clinical practice.

## 2. Materials and Methods

### 2.1. The Analysis for Differential Gene Expression and Promoter Methylation Level

The Cancer Genome Atlas (TCGA: http://ualcan.path.uab.edu/analysis.html (accessed on 21 February 2023)) [16] and Clinical Proteomic Tumor Analysis Consortium (CPTAC: http://pdc.cancer.gov/pdc/ (accessed on 26 February 2023)) [17] were used to obtained the data of mRNA levels, protein levels, and promoter methylation levels.

### 2.2. Receiver Operating Characteristic (ROC) Curve

The samples from the TCGA database were divided into two groups according to breast cancer patients and normal individuals. The software GraphPad Prism 8 (GraphPad Software Inc., San Diego, CA, USA) was used to analyze and draw ROC curves.

### 2.3. Kaplan–Meier Plotter (KM Plotter)

Based on the large databases GEO, EGA, and TCGA, we used the Kaplan–Meier plotter (http://kmplot.com/analysis/index.php?p=background (accessed on 19 May 2022)) [18] to assess the effect of genes on tumor survival.

### 2.4. Human Protein Atlas (HPA)

For analyzing the expression of DNMT1, DNMT3A, and DNMT3B, the HPA database (https://www.proteinatlas.org/ (accessed on 10 May 2023)) [19] was used to obtain the data based on immunohistochemistry assay.

### 2.5. Correlation Analysis of Gene Expression

In order to analyze the correlation of gene expression, we downloaded the expression data of the target gene from the TCGA database, and then used GraphPad Prism 8 (GraphPad Software Inc., San Diego, CA, USA) for correlation analysis and obtained the Pearson coefficient.

### 2.6. Gene Expression Omnibus (GEO)

In order to obtain the gene expression profiles of TNBC tumors and non-TNBC tumors, we downloaded relevant data from the Gene Expression Omnibus (GEO, https://www.ncbi.nlm.nih.gov/geo/ (accessed on 15 May 2023)) data set (GSE76275) and performed gene expression analysis.

### 2.7. Gene Set Enrichment Analysis (GSEA)

For gene enrichment analysis, we ranked the signature gene sets “c2.cp.kegg.v7.5.symbols.gmt” and “c5.go.v7.4.symbols.gmt” according to the expression level of TAT by GSEA 4.0.3 (Broad Institute, Cambridge, MA, USA, http://software.broadinstitute.org/gsea/index.jsp (accessed on 8 June 2023)), and divided them into TAT high-expression group and TAT low-expression group.

### 2.8. Target Overlap Analysis

Calculated and drawn custom Venn diagrams (http://bioinformatics.psb.ugent.be/webtools/Venn/ (accessed on 12 June 2023)) (provided as “free to use for all”) were used to perform the overlap analysis of target genes.

### 2.9. Molecular Docking

In order to test the potential interaction between Diazinon and target proteins (DNMT3A and DNMT3B), we used Discovery Studio (DS) v3.5 (Biovia Inc. San Diego, CA, USA) to perform molecular docking analysis. The Protein Data Bank (https://www.rcsb.org/ (accessed on 6 October 2023)) was used to obtain the crystal structures of proteins. The SDF format file of the 3D structure of Diazinon was downloaded from the PubChem database (https://pubchem.ncbi.nlm.nih.gov/ (accessed on 12 October 2023)).

### 2.10. Cell Culture

The human breast cancer cell lines MDA-MB-231 and MCF7 were cultured in DMEM supplemented with 10% (*v*/*v*) fetal bovine serum (FBS, ExCell Bio, Su Zhou, China) and 1% (*v*/*v*) penicillin/streptomycin. All cells were cultured at 37 °C in a humidified incubator containing 5% CO_2_.

### 2.11. RNA Extraction and Real-Time PCR Analysis (qPCR)

RNA was extracted using Trizol (Invitrogen, Carlsbad, CA, USA) according to the manufacturer’s protocol. LightCycler 96 (Roche, Basel, Switzerland) was used to carry out real-time PCR. The β-actin gene was used as the internal reference gene to normalize the expression level between samples. The relative expression levels between samples were calculated by the cycle threshold (2^−ΔΔCT^) method. All primer sequences of all the specific and reference genes are listed in Table 1.

### 2.12. Cell Proliferation Assay

For the cell proliferation assay, 1 × 10^4^ cells were inoculated into each well of 24-well plates. After 24 h of cell inoculation, the corresponding drugs were added to the wells to treat the cells. Then, cells were collected every 24 h after dosing, and cell counts were performed by manual counting under light microscopy using a hemocytometer. After four days, the cell survival rate was calculated and the cell growth curve was drawn.

### 2.13. Western Blot

For Western blot, treated cells were collected and fully lysed in NP-40 lysis buffer. The tissue samples were frozen in liquid nitrogen and ground with NP-40 lysis buffer. The lysates were separated in sodium dodecyl sulfate 8 to 12 % polyacrylamide gel electrophoresis and transferred onto a nitrocellulose membrane, and specific proteins were detected by enhanced chemiluminescence.

### 2.14. Colony Formation Assay

Cells were seeded in 6-well plates with 1 × 10^3^ cells per well. After 24 h, the drug was added and the culture was continued for 7–10 d. When the cell population grew to a suitable size, the cells were washed with PBS and the cell culture plates were recovered. Anhydrous methanol was carefully added to the culture wells. After the cells were fixed for 15 min, the cell masses were stained with 1 mL crystal violet. After drying, the plate was photographed and the number of cell communities in each hole was counted.

### 2.15. Reagent and Antibodies (Table 2)

The information and sources of reagents and antibodies used in this study are listed in Table 2.

**Table 2 biomolecules-14-01088-t002:** Reagent and antibodies.

Reagent or Antibody	Source	Identifier
Diazinon	TargetMol	T0998
TAT	Biodragon	BD-PE0222
DNMT3A	Selleck	F1036
DNMT3B	Selleck	F0001
γH2Ax	ABclonal	AP0687
CyclinD1	proteintech	60186-1-Ig
CDK4	proteintech	11026-1-AP
Actin	proteintech	66009-1-Ig

### 2.16. Statistics

GraphPad Prism 8 (GraphPad Software Inc., San Diego, CA, USA) was used to perform data analysis. All data were presented as mean ± standard deviation. Two-tailed Student’s *t* test was used for comparison between the two groups. The value of *p* < 0.05 was considered to be statistically significant.

## 3. Results

### 3.1. TAT Was Down-Regulated and Associated with Poor Prognosis in BC

Metabolic reprogramming is a distinct feature of tumors, and studies have found that the metabolism phenotype of essential amino acids (EAAs) contributes to the identification of TNBC subtypes [8]. More importantly, BC patients with an up-regulated EAA metabolic phenotype are more likely to exhibit chemotherapy resistance. Therefore, using amino acid metabolism as a starting point to explore biomarkers and therapeutic targets of BC is a potential approach. Moreover, we previously found that decreased tyrosine metabolism has a cancer-promoting effect in hepatocellular carcinoma and clear renal cell carcinoma, activates the cell cycle, and exacerbates malignancy [20,21]. Given the complex and diverse subtype classification of BC, we aim to explore tyrosine metabolic pathways to identify reliable biomarkers to aid in the diagnosis and therapy of BC.

First, we analyzed the expression differences of five tyrosine-metabolizing enzymes (TAT, HPD, HGD, GSTZ1, and FAH) in BC compared with normal tissues at the mRNA and protein levels, respectively. It was found that TAT and FAH are down-regulated at the mRNA and protein levels simultaneously (Figure 1A,B). To probe whether the expression of tyrosine-metabolizing enzymes possesses diagnostic significance in BC patients, the ROC curves were used to analyze the diagnostic value of tyrosine-metabolizing enzyme expression from TCGA-BC patient datasets. The ROC curve analysis showed that TAT and HGD could statistically distinguish KIRC from a normal individual, producing an area under the curve (AUC) of 0.78 (95% CI: 0.727–0.833; *p* < 0.0001) and 0.88 (95% CI: 0.843–0.920; *p* < 0.0001), respectively (Figure 1C). This result indicates that the expression of TAT and HGD has a reference value for the diagnosis of BC. Finally, we analyzed the effect of tyrosine-metabolizing enzymes on the prognostic survival of BC patients. The results showed that the low expression of TAT and GSTZ1 is significantly associated with the poor prognosis of BC patients (Figure 1D). Taking into account the expression difference, diagnostic value, and survival prognosis of tyrosine-metabolizing enzymes, we considered TAT to be a promising biomarker for BC. In fact, a previous study found that TAT may be a biomarker for BC [9], but its molecular mechanism has not been explored. Here, we will further explore the function and mechanism of TAT in BC.

### 3.2. TAT Promoter Was Methylated by DNMT3A and DNMT3B

The gene expression regulation is subject to various levels of regulation, including transcriptional level, post-transcriptional level, translational level, and post-translational level. With the deepening of research, researchers have found that epigenetics regulate gene expression by altering modifications, including DNA modification and RNA modification [22,23]. DNA modification mainly includes methylation and acetylation modification. DNA methylation modification is usually associated with transcriptional silencing [24]. To investigate this, we first analyzed the level of methylation modification on the *TAT* promoter. The analysis results showed that the promoter methylation of *TAT* is significantly increased in BC (Figure 2A), suggesting that the decreased expression of TAT might be related to DNA methylation.

In mammals, the enzymes that perform DNA methylation mainly include DNMT1 protein and DNMT3 family proteins. DNMT1 is responsible for maintaining methylation mainly during DNA replication and repair, while DNMT3 family proteins catalyze CpG de novo methylation [25,26]. We found that DNMT1, DNMT3A, and DNMT3B were all up-regulated in mRNA and protein levels in BC (Figure 2B), which further implied that the expression of *TAT* was repressed by DNA methylation. The immunohistochemistry results from the HPA database showed that DNMT3A and DNMT3B were up-regulated in BC tissues, while the expression of DNMT1 was not significantly increased (Figure 2C). Then, we analyzed the expression correlation of DNMT3A/3B and TAT in BC samples, and found that there was a significant negative correlation between the expression of DNMT3A/3B and TAT. (Figure 2D). Consistent with immunohistochemistry findings, increased DNMT3A/3B, but not DNMT1, was significantly associated with poor prognosis in BC patients (Figure 2E). The high expression of DNMTs in BC has long been reported in the literature [11], and they are also drug targets for BC [13]. Excitingly, the regulation of the tyrosine-metabolizing enzyme TAT by DNMTs was discovered for the first time. Here, we revealed that the molecular mechanism by which DNMT3A and DNMT3B, which are highly expressed in BC, methylate the promoter of *TAT* lead to the down-regulation of TAT.

To increase the reliability of our findings, we conducted validation in an independent cohort of BC patients, distinct from the cohort used in the initial analysis. The results are consistent with previous analysis, indicating a decrease in *TAT* expression in BC with significant diagnostic implications (Figure 3A,B). Notably, *DNMT3A* expression was elevated in BC, while *DNMT3B* expression remained unchanged in this cohort (Figure 3C). However, there was a significant correlation between *DNMT3A/3B* and *TAT* expression (Figure 3D). Analysis of the validation cohort suggests that the down-regulation of TAT in BC patients may be attributed to the up-regulation of DNMT3A/3B. Furthermore, we collected 12 pairs of clinical samples of BC and its corresponding normal breast tissues. We found that, compared with normal tissues, the protein levels of DNMT3A and DNMT3B in breast cancer tissues were increased, and the protein level of TAT was decreased (Figure 3E,F). Further analysis showed that TAT was negatively correlated with the protein levels of DNMT3A and DNMT3B in BC (Figure 3G).

### 3.3. DNMT3A and DNMT3B Reduce TAT Expression by Methylation of Its Promoter in Triple-Negative Breast Cancer

According to the expression of ER, PR, and HER2, BC is divided into different subtypes, among which triple-negative breast cancer (TNBC) is the most difficult subtype to treat due to its high invasiveness, metastasis, and lethality. To assess the role of TAT in TNBC, we analyzed the impact of *TAT* expression on prognostic survival in patients with receptor-positive and -negative statuses, respectively. The results showed that high TAT expression significantly improved patient survival in ER-negative, PR-positive, and HER2-negative patients, compared with the low-expression group (Figure 4A). Supposing that TAT does not work in different subtypes, the results should be consistent in receptor-negative and receptor-positive patients. However, with the present analysis of the results, we believe that TAT plays a potential functional role in patients in different subtypes.

Next, we examined TAT expression and promoter methylation levels in different subtypes of BCs. As shown in Figure 4B,C, TAT was down-regulated in different subtypes of BC compared with normal individuals, while *TAT* promoter methylation was significantly increased. To corroborate the previously elucidated regulatory mechanism, we also observed a significant increase in the expression of DNMT3A and DNMT3B across different subtypes of BC (Figure 4D), especially in TNBC. Then, we analyzed datasets that contained TNBC and non-TNBC tissue from the GEO database. The results demonstrated a significant increase in the mRNA levels of *DNMT3A* and *DNMT3B* in TNBC, accompanied by decreased *TAT* expression (Figure 4E). In addition, we also found that, with the increase in the degree of malignancy of the tumor, the expression of *TAT* gradually decreased, while the expression of DNMT3A/3B gradually increased (Figure 4F). These findings suggest that DNMT3A and DNMT3B, which are highly expressed in TNBC, methylate the promoter of *TAT*, thereby inhibiting its expression and promoting tumor progression. These findings will provide ideas for the clinical diagnosis and treatment of TNBC.

### 3.4. Reduced TAT Contributes to Activation of the Cell Cycle and Promotes DNA Repair

To elucidate the biological function of low-expressed TAT in BC, we categorized BC patients into high- and low-*TAT* expression groups and conducted pathway enrichment analysis. Bioinformatics analysis revealed that low expression of TAT promoted sister chromatid segregation, DNA replication, and cell cycle (Figure 5A). The effects of tyrosine metabolic pathways on the cell cycle have also been found in hepatocellular carcinoma and renal clear cell carcinoma [20,21], suggesting a common regulatory mechanism in tumors. Upon overlapping related genes in the three pathways, we found that *CDC6* may be a downstream target gene affected by TAT (Figure 5B), and high expression of *CDC6* in BC was associated with the poor prognosis of patients (Figure 5C,D). In addition, we observed that a low expression of *TAT* also contributed to DNA repair, especially homologous recombination repair and mismatch repair (Figure 5E). Rapid growth and proliferation are hallmark characteristics of tumor cells, necessitating large-scale DNA synthesis for distribution to progeny cells. Intense replication stress induces DNA damage during and after replication. In both tumor cells and normal cells, if the DNA damage cannot be repaired in time, it may cause fatal damage to the cell. We also identified four DNA repair-related target genes, namely *RPA3*, *RPA4*, *POLD1*, and *POLD2* (Figure 5F). In addition to RPA4, the expression of RPA3, POLD1, and POLD2 was significantly increased in BC (Figure 5G). Taken together, these results indicate that reduced TAT expression not only contributes to cell cycle activation but also enhances DNA damage repair capacity to ensure orderly DNA replication.

### 3.5. Targeting DNMT3A/3B Inhibits Proliferation of Breast Cancer Cells

Our above results showed that TAT was down-regulated as a tumor suppressor in TNBC. We previously found that tyrosine-metabolizing enzymes, also acting as tumor suppressors, are inhibited in liver cancer, and we identified that Diazinon could up-regulate the expression of metabolic enzymes as an activator [21]. Here, we speculate whether Diazinon inhibits the activity of DNMT3A/B, thereby up-regulating *TAT* expression and exerting an inhibitory effect on BC. To investigate this, we analyzed the interaction between Diazinon and DNMT3A/3B using molecular docking software. The results revealed that Diazinon directly binds to DNMT3A and DNMT3B (Figure 6A), suggesting its potential ability to inhibit DNMT3A/B.

In order to prove the anti-tumor effect of Diazinon in BC, we used it to treat breast cancer cells and observed that it increased the expression of TAT while suppressing the expression of *CDC6*, *RPA3*, *POLD1*, and *POLD2* (Figure 6B). Moreover, we also found that Diazinon promoted DNA damage by detecting γH2Ax and inhibited cell cycle by detecting CyclinD1 and CDK4 (Figure 6C). In addition, the colony formation and proliferation of tumor cells were significantly inhibited by Diazinon (Figure 6D–G). These findings indicate that the expression of TAT is regulated by DNMT3A/B in BC, which mediates the downstream cell cycle and DNA repair process, thereby exerting a tumor suppressor effect. TAT also shows promise as a potential biomarker. Furthermore, Diazinon up-regulates *TAT* expression by targeting DNMT3A and DNMT3B, which provides a new clinically significant potential treatment strategy for BC.

## 4. Discussion

BC remains a public health problem worldwide. Over the past few decades, advances in BC research have led to a greater understanding of the disease, positively impacting its diagnosis and treatment. BC is a heterogeneous tumor, with different subtypes exhibiting distinct pathological features and tumor properties [27,28]. The purpose of BC classification is to provide reference for accurate tumor diagnosis and effective treatment decisions, and to escort accurate treatment. Traditional BC classification mainly relies on clinicopathological features and the evaluation of conventional biomarkers, which may have certain limitations. However, in this study, we adopted a novel approach by focusing on tumor metabolic reprogramming, specifically amino acid metabolism. This perspective enables us to discover new metabolic biomarkers and provides new insights into the development of new targets for precise treatment.

It has been reported that tumor metabolic reprogramming (including glucose metabolism, fatty acid metabolism, and amino acid metabolism) plays a crucial role in BC metastasis and chemotherapy resistance [29]. For example, the metabolic enzymes of glycolysis are highly expressed in BC, promoting glycolysis and tumor progression [30]. In addition, fatty acid synthesis, oxidation, uptake, and storage play important roles in the progression of TNBC [31]. Among the amino acid metabolism pathways in BC, glutamine, serine, and cysteine have been extensively studied [32,33,34]. Glutamine is not only involved in biosynthesis but also maintains redox homeostasis, providing advantages for tumor cell proliferation, invasion, and metastasis. There is evidence that the reduction of serine and glycine in the environment inhibits the growth of tumor cells [35]. This shows how important metabolic reprogramming is for the occurrence and development of BC. To identify effective biomarkers in BC, we analyzed the tyrosine metabolic pathway and identified TAT to be a promising target. Consistent with previous reports [9], *TAT* expression was decreased in BC. However, the specific molecular mechanism is still unclear. So, we further explored the role and biological function of TAT in BC. We found that *TAT* expression was significantly suppressed in BC and closely associated with the poor prognosis in BC patients. Reduced TAT levels could promote DNA replication, activate the cell cycle, and facilitate cell proliferation. In fact, our previous work also found an inextricable link between tyrosine metabolism and abnormal activation of the cell cycle [20,21], which suggests that the correlation between tyrosine metabolism pathways and the cell cycle is a common mechanism in tumors. In addition, we also found that TAT can promote DNA damage repair, especially mismatch repair during replication. It is well known that the rapid proliferation of tumor cells requires a large amount of DNA replication, which inevitably generates DNA damage. And DNA damage activates the cell cycle checkpoints, temporarily halting the cell cycle. Therefore, the reduction of TAT not only promotes DNA replication but also improves the ability of DNA damage repair, ensuring the orderly progression of the cell cycle.

To elucidate the molecular mechanism of TAT low expression, we found that the TAT promoter is hypermethylated in BC. It is well known that DNA methylation is associated with gene silencing. The high expression of DNA methyltransferases (DNMTs) in BC further confirmed our suspicion. Most importantly, the expression of TAT was significantly decreased in tumor cells with a high expression of DNMT3A and DNMT3B, and there was an obvious negative correlation between TAT and DNMT3A/3B. DNMTs have long been known to play a tumor-promoting role in BC, and are also important targets for drug therapy [13]. However, this is the first time it has been shown that DNMT3A and DNMT3B mediate DNA methylation to regulate the expression of tyrosine-metabolizing enzyme TAT, which affects the progression of BC. Our study enriches the understanding of the role of DNMT3A and DNMT3B in promoting BC progression and provides new ideas and targets for BC treatment. Not only that, we also found that the regulatory axis of DNMT3A/3B-TAT is applicable in triple-negative breast cancer (TNBC). This finding suggests that targeting this regulatory axis contributes to the development of new drugs and therapeutic strategies for TNBC. In conclusion, our study enriches the understanding of the role of DNMTs in promoting BC progression and provides new ideas and targets for TNBC treatment.

We revealed that the tyrosine metabolic pathway plays a role in BC inhibition. In our previous work, we found that Diazinon up-regulates the expression of tyrosine-metabolizing enzymes in hepatocellular carcinoma [21]. Therefore, we speculated whether Diazinon could up-regulate the expression of TAT in BC and play a tumor suppressor role by inhibiting DNMT3A/3B. Excitingly, molecular docking results showed that Diazinon could bind to DNMT3A and DNMT3B. Further experiments confirmed the inhibition of Diazinon on the growth of BC cells and the regulation of downstream target expression. In general, we identified Diazinon as a small molecule inhibitor targeting DNMT3A/3B, which up-regulates *TAT* expression, blocks the cell cycle in BC cells, and exerts tumor suppressor function.

## 5. Conclusions

The external manifestation of tumor heterogeneity is that different subtypes of tumors have distinct pathological characteristics and varying sensitivities to the same treatment. BC, which seriously threatens women’s health, encompasses a variety of subtypes, posing challenges for its diagnosis and treatment. To address this, we focused on amino acid metabolism and screened and identified tyrosine aminotransferase (TAT) as a novel biomarker for BC. Mechanismly, we confirmed the regulation of *TAT* expression by DNMT3A and DNMT3B, which enriched our understanding of epigenetics. Furthermore, we clarified the involvement of TAT in cell cycle and DNA damage repair. These findings provide new insights for future scientific research and new strategies for clinical treatment of BC. However, it is undeniable that the BC biomarker and related genes require further experimental validation on the basis of a rigorous attitude.

## Figures and Tables

**Figure 1 biomolecules-14-01088-f001:**
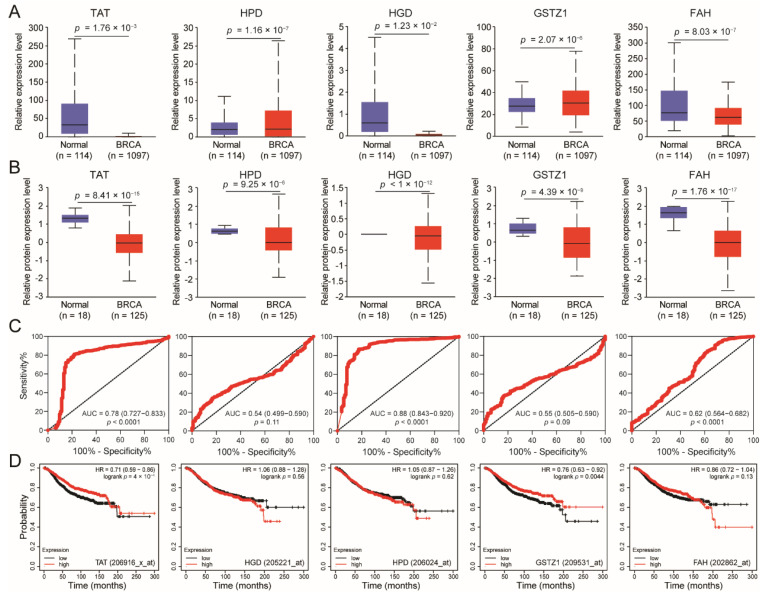
TAT is down-regulated and associated with poor prognosis in BC. (**A**,**B**) Tyrosine-metabolizing enzymes (TAT, HPD, HGD, GSTZ1, and FAH) and mRNA and protein expression levels between tumor and normal tissues in patients with BC in TCGA database. (**C**) ROC curve analysis for diagnostic value of tyrosine-metabolizing enzymes (TAT, HPD, HGD, GSTZ1, and FAH) in BC. AUC—area under curve. (**D**) Overall survival of patients with BC grouped by tyrosine-metabolizing enzyme expression through the Kaplan-Meier plotter online analysis tool. All error bars, mean values ± SD, and *p* values were determined by unpaired two-tailed Student’s *t* test or one way ANOVA of *n* = 3 independent biological experiments.

**Figure 2 biomolecules-14-01088-f002:**
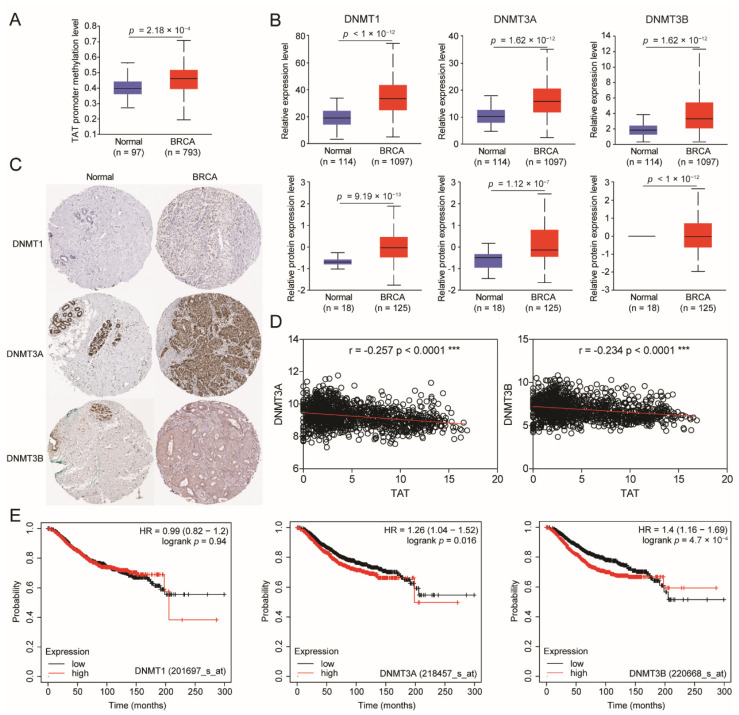
*TAT* promoter is methylated by DNMT3A and DNMT3B. (**A**) Methylation level of *TAT* promoter between tumor and normal tissues in patients with BC in TCGA database. (**B**) DNA methyltransferases (DNMT1, DNMT3A, and DNMT3B) and mRNA and protein expression levels between tumor and normal tissues in patients with BC in TCGA database. (**C**) Representative IHC staining of DNMT1, DNMT3A, and DNMT3B in BC from HPA database. (**D**) Expression correlation between DNMT3A/B and TAT in BC patients. Each black circle represents the level of two proteins in each sample. The red lines represent the linear relationship between the two proteins in these samples. (**E**) Overall survival of patients with BC grouped by DNMT1/DNMT3A/DNMT3B expression through the Kaplan–Meier plotter online analysis tool. All error bars, mean values ± SD, and *p* values were determined by unpaired two-tailed Student’s *t* test or one way ANOVA of *n* = 3 independent biological experiments. *** *p* < 0.001.

**Figure 3 biomolecules-14-01088-f003:**
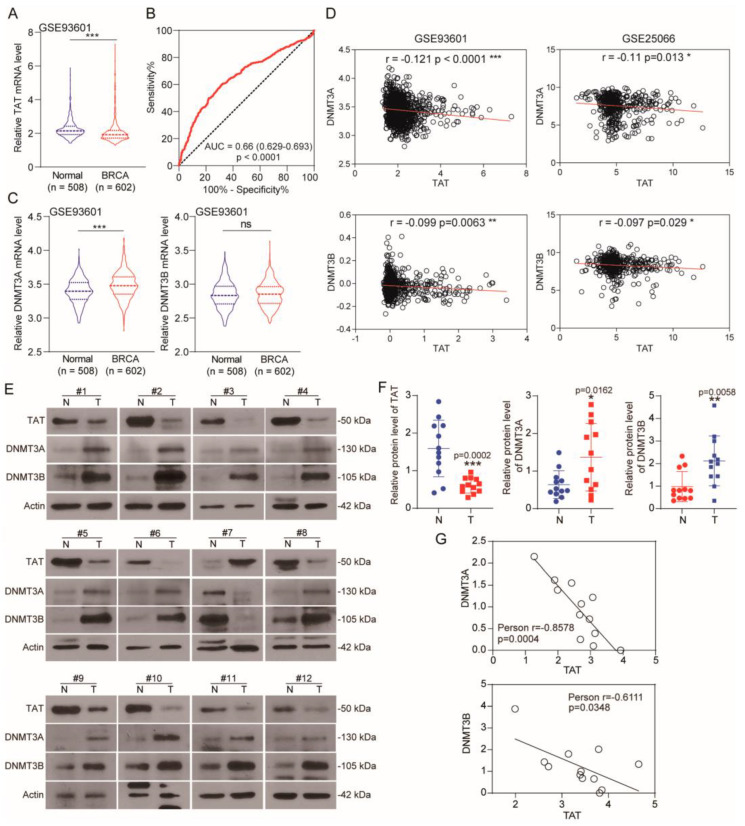
Expression of TAT and DAMT3A/3B was negatively correlated in TNBC. (**A**) mRNA level of *TAT* between tumor and normal tissues in patients with BC in TCGA database. (**B**) ROC curve analysis for diagnostic value of TAT in GSE93601. (**C**) mRNA level of *DNMT3A/3B* between tumor and normal tissues in patients with BC in TCGA database. (**D**) Expression correlation between DNMT3A/B and TAT. Each black circle represents the level of two proteins in each sample. The red lines represent the linear relationship between the two proteins in these samples. (**E**) Expression of TAT, DNMT3A, and NDMT3B in the paired tumor-adjacent normal breast tissues (N) and human BC tissues (T) by WB. (**F**) Levels of TAT, DNMT3A, and DNMT3B were analyzed in clinical normal breast tissues and BC tissues. (**G**) The correlation between TAT and DNMT3A/DNMT3B protein expression in clinical samples were analyzed. Each black circle represents the level of two proteins in each sample. The red lines represent the linear relationship between the two proteins in these samples. All error bars, mean values ± SD, and *p* values were determined by unpaired two-tailed Student’s *t* test or one way ANOVA of *n* = 3 independent biological experiments. * *p* < 0.05; ** *p* < 0.01; *** *p* < 0.001. Original images of (**E**) can be found in Appendix A.

**Figure 4 biomolecules-14-01088-f004:**
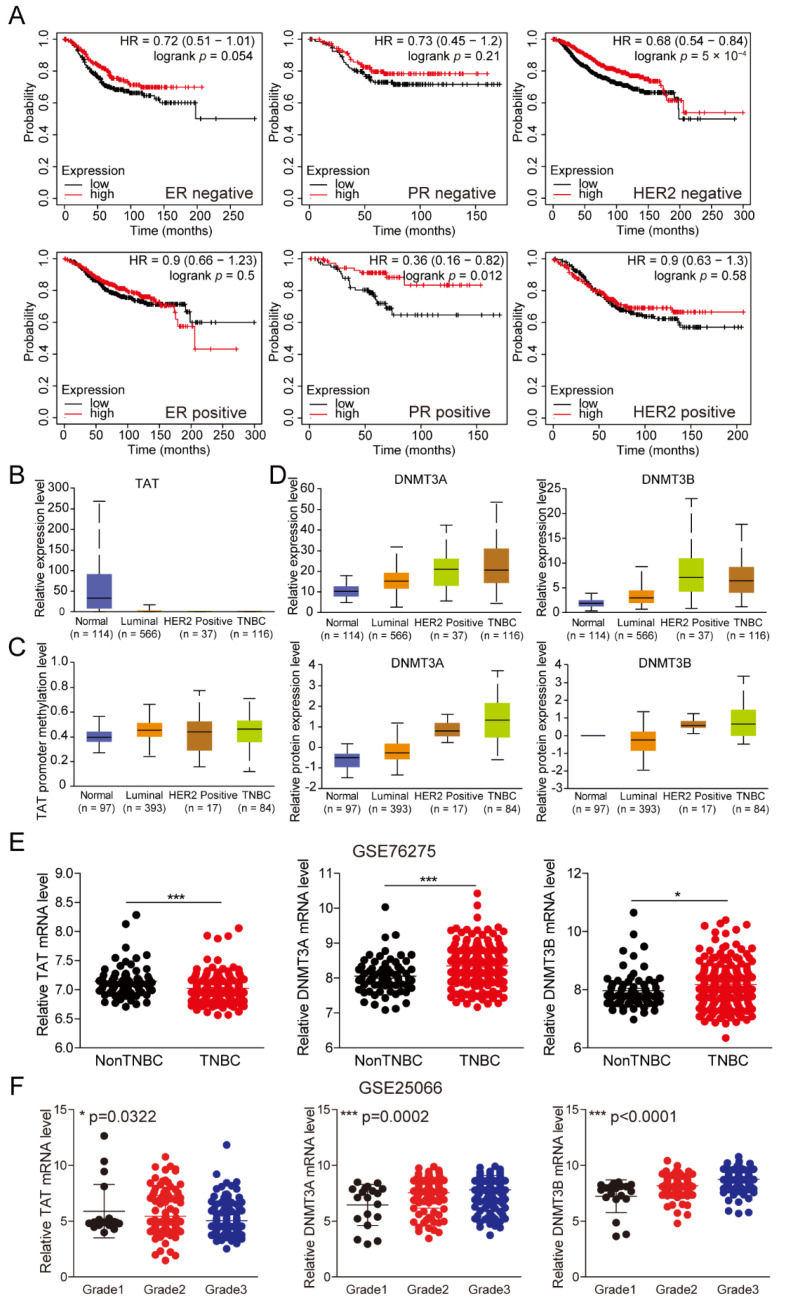
DNMT3A and DNMT3B methylate the promoter resulting in decreased TAT expression in TNBC. (**A**) Overall survival of BC patients with different receptor expression levels grouped by TAT expression using the Kaplan–Meier plotter online analysis tool. (**B**) *TAT* mRNA expression level between different subclasses of tumor and normal tissues in patients with BC in TCGA database. (**C**) *TAT* promoter methylation level between different subclasses of tumor and normal tissues in patients with BC in TCGA database. (**D**) *DNMT3A/3B* mRNA and protein expression levels between different subclasses of tumor and normal tissues in patients with BC in TCGA database. (**E**) mRNA level of *DNMT3A/3B* and *TAT* between TNBC tumor and non-TNBC tumor tissues in patients with BC in GEO database. (**F**) mRNA level of *DNMT3A/3B* and *TAT* in the tumor tissues of BC patients with different malignant degrees in GEO database. All error bars, mean values ± SD, and *p* values were determined by unpaired two-tailed Student’s *t* test or one way ANOVA of *n* = 3 independent biological experiments. * *p* < 0.05; *** *p* < 0.001.

**Figure 5 biomolecules-14-01088-f005:**
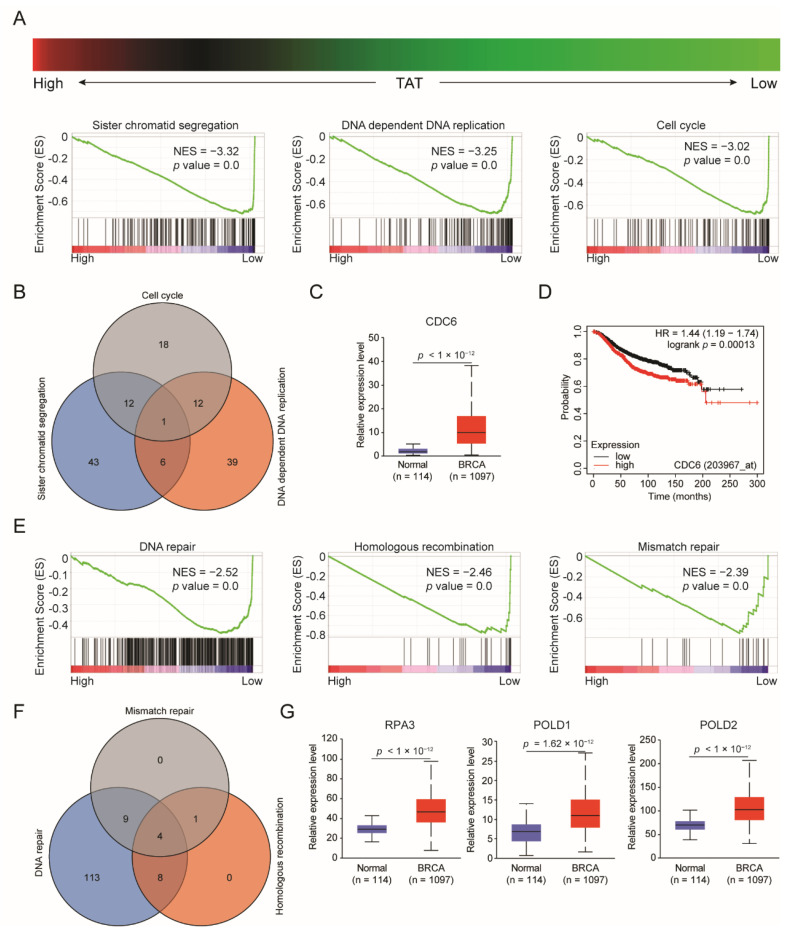
Decreased TAT promotes activation of the cell cycle and promotes DNA repair. (**A**) GSEA pathway enrichment analyses of TAT signature in patients with BC from the TCGA datasets. (**B**) Overlapping analysis for related genes of sister chromatid segregation, DNA-dependent DNA replication, and cell cycle. (**C**) *CDC6* mRNA expression levels between tumor and normal tissues in patients with BC in the TCGA database. (**D**) Overall survival of patients with BC grouped by *CDC6* expression through the Kaplan–Meier plotter online analysis tool. (**E**) GSEA pathway enrichment analyses of TAT signature in patients with BC from the TCGA datasets. (**F**) Overlapping analysis for related genes of DNA repair, homologous recombination, and mismatch repair. (**G**) *RPA3*, *POLD1*, and *POLD2* mRNA expression levels between tumor and normal tissues in patients with BC in the TCGA database. All error bars, mean values ± SD, and *p* values were determined by unpaired two-tailed Student’s *t* test or one way ANOVA of *n* = 3 independent biological experiments.

**Figure 6 biomolecules-14-01088-f006:**
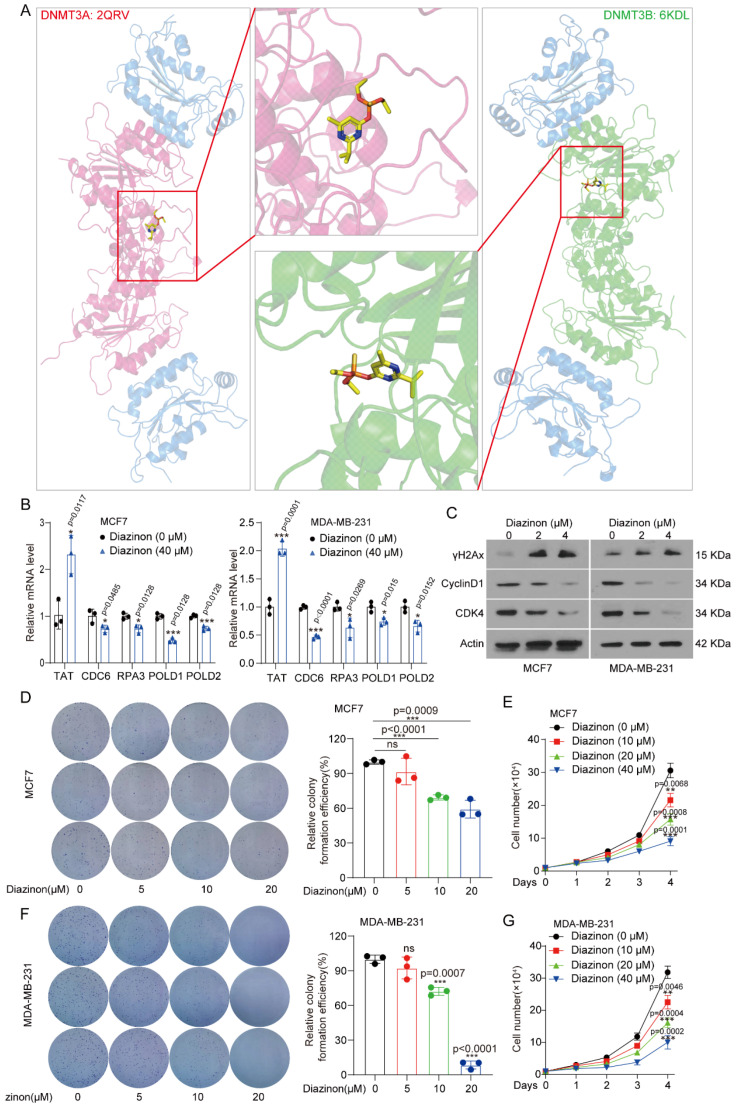
Targeting DNMT3A/3B inhibits growth and proliferation of breast cancer cells. (**A**) Molecular models of Diazinon binding to DNMT3A/DNMT3B. (**B**) Expression of *TAT*, *RPA3*, *POLD1*, and *POLD2* in MCF7 and MDA-MB-231 cells with the treatment of Diazinon by qPCR. (**C**) The protein levels of γH2Ax, CyclinD1, and CDK4 in MDA-MB-231 cells with the treatment of Diazinon by qPCR. (**D**) Colony formation of MCF7 cells with the treatment of Diazinon. (**E**) Proliferation of MCF7 cells with the treatment of Diazinon. (**F**) Colony formation of MDA-MB-231 cells with the treatment of Diazinon. (**G**) Proliferation of MDA-MB-231 cells with the treatment of Diazinon. All error bars, mean values ± SD, and *p* values were determined by unpaired two-tailed Student’s *t* test or one way ANOVA of *n* = 3 independent biological experiments. * *p* < 0.05; ** *p* < 0.01; *** *p* < 0.001. Original images of (**C**) can be found in Appendix A.

**Table 1 biomolecules-14-01088-t001:** Sequence of primers for real-time PCR.

Gene	Sequence (5′→3′)
*TAT*	Primer-F: CTGGACTCGGGCAAATATAATGG Primer-R: GTCCTTAGCTTCTAGGGGTGC
*CDC6*	Primer-F: CCAGGCACAGGCTACAATCAG Primer-R: AACAGGTTACGGTTTGGACATT
*RPA3*	Primer-F: AGCTCAATTCATCGACAAGCC Primer-R: TCTTCATCAAGGGGTTCCATCA
*POLD1*	Primer-F: CAGTGCCAAGGTGGTGTATGG Primer-R: CTTGCTGATAAGCAGGTATGGG
*POLD2*	Primer-F: CAGCGTATCAAACTAAAAGGCAC Primer-R: GTCTCTCACGGAGCCAAACAC
*ACTIN*	Primer-F: GGAAATCGTGCGTGACAT Primer-R: TGCCAATGGTGATGACCT

## Data Availability

All data will be made available upon reasonable request by emailing the corresponding author.

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
