# Peer review of "Identification of TAT as a Biomarker Involved in Cell Cycle and DNA Repair in Breast Cancer"

_biomolecules, 2024, doi:10.3390/biom14091088_

Round 1

Reviewer 1 Report

Comments and Suggestions for Authors

In this study, Xie et al. identified tyrosine aminotransferase (TAT) as a biomarker involved in cell cycle and DNA repair in breast cancer. Treatment of triple-negative breast cancer (TNBC) remains particularly challenging due to its resistance to chemotherapy and poor prognosis. Therefore, identifying reliable biomarkers for TNBC is of great clinical importance. This finding provides insights into the molecular typing of breast cancer and the development of new treatment strategies in clinical practice. However, this article is not suitable for publication in its current form.

Firstly, the novelty of this work is questionable as the tyrosine metabolizing enzyme TAT was previously identified as a breast cancer biomarker by computational methods by Ansar et al., 2019 (PMID: 31703592).

The authors claim that they explored the biological function and molecular mechanism of TAT in breast cancer. However, they analyzed preexisting datasets and performed bioinformatic analysis without validating their findings through molecular and cell biology techniques. For example, in Figure 3, the expression of TAT and DAMT3A/3B was negatively correlated in TNBC. This observation requires validation at the protein level. The authors could perform Western blot analysis to validate the correlation of TAT and DAMT3A/3B in tumor and normal tissues in patients with BRCA.

Reviewer 2 Report

Comments and Suggestions for Authors

I found the work that you have been doing to be very interesting. I would like to make the following suggestions to improve the quality of the work:

* Throughout the text, breast cancer should be changed to BRCA whenever possible.

* In line 54 the meaning of the acronym should be indicated and in line 64 only the acronym should be left.

* Line 57 should contain the acronym TNBC.  Check the text to see if it should be elsewhere in the article.

* The paragraph from line 64 to 76 would be the conclusion of the article.  In this sense, I would include a paragraph about the importance of the TAT study and diazinon in BRCA.  And if you see fit, you could rewrite the conclusion if you decide to include this paragraph, as I find it very interesting and it defines your study very well.

* In line 184 I would delete the full stop after the word cancer.

* The sentence from line 348 to 350 should be rewritten. The word diazinon is repeated several times.
